# FAIR MIXUP: FAIRNESS VIA INTERPOLATION

**Ching-Yao Chuang**[*]
CSAIL, MIT
cychuang@mit.edu

**Youssef Mroueh**
IBM Research AI
mroueh@us.ibm.com

## ABSTRACT

Training classifiers under fairness constraints such as group fairness, regularizes the disparities of predictions between the groups. Nevertheless, even though the constraints are satisfied during training, they might not generalize at evaluation time. To improve the generalizability of fair classifiers, we propose *fair mixup*, a new data augmentation strategy for imposing the fairness constraint. In particular, we show that fairness can be achieved by regularizing the models on paths of interpolated samples between the groups. We use *mixup*, a powerful data augmentation strategy to generate these interpolates. We analyze *fair mixup* and empirically show that it ensures a better generalization for both accuracy and fairness measurement in tabular, vision, and language benchmarks. The code is available at https://github.com/chingyaoc/fair-mixup.

## 1 INTRODUCTION

Fairness has increasingly received attention in machine learning, with the aim of mitigating unjustified bias in learned models. Various statistical metrics were proposed to measure the disparities of model outputs and performance when conditioned on sensitive attributes such as gender or race. Equipped with these metrics, one can formulate constrained optimization problems to impose fairness as a constraint. Nevertheless, these constraints do not necessarily generalize since they are *data-dependent*, i.e they are estimated from finite samples. In particular, models that minimize the disparities on training sets do not necessarily achieve the same fairness metric on testing sets (Cotter et al., 2019). Conventionally, regularization is required to improve the generalization ability of a model (Zhang et al., 2016). On one hand, explicit regularization such as weight decay and dropout constrain the model capacity. On the other hand, implicit regularization such as data augmentation enlarge the support of the training distribution via prior knowledge (Hernández-García & König, 2018).

In this work, we propose a data augmentation strategy for optimizing group fairness constraints such as *demographic parity* (DP) and *equalized odds* (EO) (Barocas et al., 2019). Given two sensitive groups such as male and female, instead of directly restricting the disparity, we propose to regularize the model on interpolated distributions between them. Those augmented distributions form a *path* connecting the two sensitive groups. Figure 1 provides an illustrative example of the idea. The path simulates how the distribution transitions from one group to another via interpolation. Ideally, if the model is invariant to the sensitive attribute, the expected prediction of the model along the path should have a smooth behavior. Therefore, we propose a regularization that favors smooth transitions along the path, which provides a stronger prior on the model class.

We adopt *mixup* (Zhang et al., 2018b), a powerful data augmentation strategy, to construct the interpolated samples. Owing to mixup's simple form, the smoothness regularization we introduce has a closed form expression that can be easily optimized. One disadvantage of mixup is that the interpolated samples might not lie on the natural data manifold. Verma et al. (2019) propose *Manifold Mixup*, which generate the mixup samples in a latent space. Previous works (Bojanowski et al., 2018; Berthelot et al., 2018) have shown that interpolations between a pair of latent features correspond to semantically meaningful, smooth interpolation in the input space. By constructing the path in the latent space, we can better capture the semantic changes while traveling between the sensitive groups and hence result in a better fairness regularizer that we coin *fair mixup*. Empirically, fair

---

[*]Work done during an internship at IBM Research AI

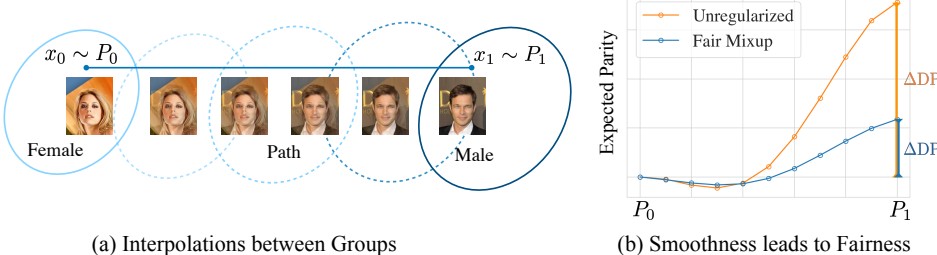

(a) Interpolations between Groups

(b) Smoothness leads to Fairness

Figure 1: (a) Visualization of the path constructed via mixup interpolations between groups that have distribution $P_0$ and $P_1$, respectively. (b) Fair mixup penalizes the changes in model's expected prediction with respect to the interpolated distributions. The regularized model (blue curve) has smaller slopes comparing to the unregularized one (orange curve) along the path from $P_0$ to $P_1$, which eventually leads to smaller demographic parity $\Delta$DP.

mixup improves the generalizability for both DP and EO on tabular, computer- vision, and natural language benchmarks. Theoretically, we prove for a particular case that fair mixup corresponds to a Mahalanobis metric in the feature space in which we perform the classification. This metric ensures group fairness of the model, and involves the Jacobian of the feature map as we travel along the path.

In short, this work makes the following contributions:

- We develop *fair mixup*, a data augmentation strategy that improves the generalization of group fairness metrics;
- We provide a theoretical analysis to deepen our understanding of the proposed method;
- We evaluate our approach via experiments on tabular, vision, and language benchmarks;

## 2  RELATED WORK

**Machine Learning Fairness**   To mitigate unjustified bias in machine learning systems, various fairness definitions have been proposed. The definitions can usually be classified into *individual fairness* or *group fairness*. A system that is individually fair will treat similar users similarly, where the similarity between individuals can be obtained via prior knowledge or metric learning (Dwork et al., 2012; Yurochkin et al., 2019). Group fairness metrics measure the statistical parity between subgroups defined by the sensitive attributes such as gender or race (Zemel et al., 2013; Louizos et al., 2015; Hardt et al., 2016). While fairness can be achieved via pre- or post-processing, optimizing fair metrics at training time can lead to the highest utility (Barocas et al., 2019). For instance, Woodworth et al. (2017) impose independence via regularizing the covariance between predictions and sensitive attributes. Zafar et al. (2017) regularize decision boundaries of convex margin-based classifier to minimize the disparaty between groups. Zhang et al. (2018a) mitigate the bias via minimizing an adversary's ability to predict sensitive attributes from predictions.

Nevertheless, these constraints are data-dependent, even though the constraints are satisfied during training, the model may behave differently at evaluation time. Agarwal et al. (2018) analyze the generalization error of fair classifiers obtained via two-player games. To improve the generalizability, Cotter et al. (2019) inherit the two-player setting while training each player on two separated datasets. In spite of the analytical solutions and theoretical guarantees, game-theoretic approaches could be hard to scale for complex model classes. In contrast, our proposed fair mixup, is a general data augmentation strategy for optimizing the fairness constraints, which is easily compatible with any dataset modality or model class.

**Data Augmentation and Regularization**   Data augmentation expands the training data with examples generated via prior knowledge, which can be seen as an implicit regularization (Zhang et al., 2016; Hernández-García & König, 2018) where the prior is specified as virtual examples. Zhang et al. (2018b) proposes *mixup*, which generate augmented samples via convex combinations of pairs of examples. In particular, given two examples $z_i, z_j \in \mathbb{R}^d$ where $z$ could include both input and label, mixup constructs virtual samples as $tz_i + (1 - t)z_j$ for $t \in [0, 1]$. State-of-the-art results

are obtained via training on mixup samples in different modalities. Verma et al. (2019) introduces manifold mixup and shows that performing mixup in a latent space further improves the generalization. While previous works focus on general learning scenarios, we show that regularizing models on mixup samples can lead to group fairness and improve generalization.

## 3 GROUP FAIRNESS

Without loss of generality, we consider the standard fair binary classification setup where we obtain inputs $X \in \mathcal{X} \subset \mathbb{R}^d$, labels $Y \in \mathcal{Y} = \{0, 1\}$, sensitive attribute $A \in \{0, 1\}$, and prediction score $\hat{Y} \in [0, 1]$ from model $f : \mathbb{R}^d \rightarrow [0, 1]$. We will focus on demographic parity (DP) and equalized odds (EO) in this work, while our approach also encompasses other fairness metrics (detailed discussion in section 5). DP requires the predictions $\hat{Y}$ to be independent of the sensitive attribute $A$, that is, $P(\hat{Y}|A = 0) = P(\hat{Y}|A = 1)$. However, DP ignores the possible correlations between $Y$ and $A$ and could rule out the perfect predictor if $Y \not\perp A$. EO overcomes the limit of DP by conditioning on the label $Y$. In particular, EO requires $\hat{Y}$ and $A$ to be conditionally independent with respect to $Y$, that is, $P(\hat{Y}|A = 1, Y = y) = P(\hat{Y}|A = 0, Y = y)$ for $y \in \{0, 1\}$. Given the difficulty of optimizing the independency constraints, Madras et al. (2018) propose the following relaxed metrics:

$$\Delta\text{DP}(f) = |\mathbb{E}_{x \sim P_0} f(x) - \mathbb{E}_{x \sim P_1} f(x)| \quad \Delta\text{EO}(f) = \sum_{y \in \{0,1\}} \left| \mathbb{E}_{x \sim P_0^y} f(x) - \mathbb{E}_{x \sim P_1^y} f(x) \right|$$

where we define $P_a = P(\cdot|A = a)$ and $P_a^y = P(\cdot|A = a, Y = y)$, $a, y \in \{0, 1\}$. We denote the joint distribution of $X$ and $Y$ by $P$. Similar metrics have also been used in Agarwal et al. (2018), Wei et al. (2019), and Taskesen et al. (2020). One can formulate a penalized optimization problem to regularize the fairness measurement, for instance,

$$\text{(Gap Regularization):} \quad \min_f \mathbb{E}_{(x,y) \sim P}[\ell(f(x), y)] + \lambda\Delta\text{DP}(f), \tag{1}$$

where $\ell$ is the classification loss. In spite of its simplicity, our experiments show that small training values of $\Delta\text{DP}(f)$ do not necessarily generalize well at evaluation time (See section 6). To improve the generalizability, we introduce a data augmentation strategy via a dynamic form of group fairness metrics.

## 4 DYNAMIC FORMULATION OF FAIRNESS: PATHS BETWEEN GROUPS

For simplicity, we will first consider $\Delta\text{DP}$ as the fairness metric, and extend our development to $\Delta\text{EO}$ in section 5. $\Delta\text{DP}$ provides a static measurement by quantifying the expected difference at $P_0$ and $P_1$. In contrast, one can consider a dynamic metric that measures the change of $\hat{Y}$ while transitioning gradually from $P_0$ to $P_1$. To convert from the static to the dynamic formulations, we start with a simple Lemma that bridges two groups with an interpolator $T(x_0, x_1, t)$, which generates interpolated samples between $x_0$ and $x_1$ based on step $t$.

**Lemma 1.** *Let $T : \mathcal{X}^2 \times [0, 1] \rightarrow \mathcal{X}$ be a function continuously differentiable w.r.t. $t$ such that $T(x_0, x_1, 0) = x_0$ and $T(x_0, x_1, 1) = x_1$. For any differentiable function $f$, we have*

$$\Delta\text{DP}(f) = \left| \int_0^1 \frac{d}{dt} \int f(\underbrace{T(x_0, x_1, t)}_{\text{interpolation}}) dP_0(x_0) dP_1(x_1) dt \right| =: \left| \int_0^1 \frac{d}{dt} \mu_f(t) dt \right|, \tag{2}$$

*where we define $\mu_f(t) = \mathbb{E}_{x_0 \sim P_0, x_1 \sim P_1} f(T(x_0, x_1, t))$, the expected output of $f$ with respect to $T(x_0, x_1, t)$.*

Figure 2 provides an illustrative example of the idea. Lemma 1 relaxes the binary sensitive attribute into a continuous variable $t \in [0, 1]$, where $\mu_f$ captures the behavior of $f$ while traveling from group 0 to group 1 along the path constructed with the interpolator $T$. In particular, given two examples $x_0$ and $x_1$ drawn from each group, $T$ generates interpolated samples that change smoothly with respect to $t$.

For instance, given two racial backgrounds in the dataset, $\mu_f$ simulates how the prediction of $f$ changes while the data of one group smoothly transforms to another. We can then detect whether there are "unfair" changes in $\mu_f$ along the path. The dynamic formulation allows us to measure the sensitivity of $f$ with respect to a relaxed continuous sensitive attribute $t$ via the derivative $\frac{d}{dt}\mu_f(t)$. Ideally, if $f$ is invariant to the sensitive attribute, $\frac{d}{dt}\mu_f(t)$ should be small along the path from $t = 0$ to $1$. Importantly, a small $\Delta$DP does not imply $|\frac{d}{dt}\mu_f(t)|$ is small for $t \in [0, 1]$ since the derivative could fluctuate as it can be seen in Figure 2.

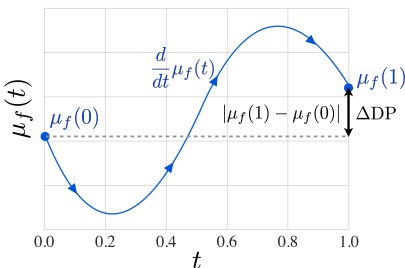

Figure 2: The expected output $\mu_f(t)$ gradually changes as $t \to 1$. Even when $\Delta$DP is small, $|\frac{d}{dt}\mu_f(t)|$ could still be large along the path.

### 4.1 SMOOTHNESS REGULARIZATION

To make $f$ invariant to $t$, we propose to regularize the derivative along the path:

$$\text{(Smoothness Regularizer):} \quad R_T(f) = \int_0^1 \left| \frac{d}{dt}\mu_f(t) \right| dt. \tag{3}$$

Interestingly, $R_T(f)$ is the arc length of the curve defined by $\mu_f(t)$ for $t \in [0, 1]$. Now, we can interpret the problem from a geometric point of view. The interpolator $T$ defines a curve $\mu_f(t) :$ $[0, 1] \to \mathbb{R}$, and $\Delta\text{DP}(f) = |\mu_f(0) - \mu_f(1)|$ is the Euclidean distance between points $t = 0$ and $1$. $\Delta\text{DP}(f)$ fails to capture the behavior of $f$ while transitioning from $P_0$ to $P_1$. In contrast, regularizing the arc length $R_T(f)$ favors a smooth transition from $t = 0$ to $1$, which constrains the fluctuation of the function as the sensitive attributes change. By Jensen's inequality, $\Delta\text{DP}(f) \leq R_T(f)$ for any $f$, which further justifies the validity of regularizing $\Delta\text{DP}(f)$ with $R_T(f)$.

## 5 FAIR MIXUP: REGULARIZING MIXUP PATHS

It remains to determine the interpolator $T$. A good interpolater shall (1) generate meaningful interpolations, and (2) the derivative of $\mu_f(.)$ with respect to $t$ should be easy to compute. In this section, we show that mixup (Zhang et al., 2018b), a powerful data augmentation strategy, is itself a valid interpolator that satisfies both criterions.

**Input Mixup** We first adopt the standard mixup (Zhang et al., 2018b) by setting the interpolator as the linear interpolation in input space: $T(x_0, x_1, t) = tx_0 + (1 - t)x_1$. It can be verified that $T_{\text{mixup}}$ satisfies the interpolator criterion defined in Lemma 1. The resulting smoothness regularizer has the following closed form expression[1]:

$$R_{\text{mixup}}^{\text{DP}}(f) = \int_0^1 \left| \int \langle \nabla_x f(tx_0 + (1-t)x_1), x_0 - x_1 \rangle \, dP_0(x_0) dP_1(x_1) \right| dt.$$

The regularizer can be easily optimized by computing the Jacobian of $f$ on mixup samples. Jacobian regularization is a common approach to regularize neural networks (Drucker & LeCun, 1992). For instance, regularizing the norm of the Jacobian can improve adversarial robustness (Chan et al., 2019; Hoffman et al., 2019). Here, we regularize the expected inner product between the Jacobian on mixup samples and the difference $x_0 - x_1$.

**Manifold Mixup** One disadvantage of input mixup is that the curve is defined with mixup samples, which might not lie on the natural data manifold. Verma et al. (2019) propose *Manifold Mixup*, which generate the mixup samples in the latent space $\mathcal{Z}$. In particular, manifold mixup assumes a compositional hypothesis $f \circ g$ where $g : \mathcal{X} \to \mathcal{Z}$ is the feature encoder and the predictor $f : \mathcal{Z} \to \mathcal{Y}$ takes the encoded feature to perform prediction. Similarly, we can establish the equivalence between

---

[1]exchange the derivative and integral via the Dominated Convergence Theorem

$\Delta$DP and manifold mixup:

$$\Delta\text{DP}(f \circ g) = \left| \int_0^1 \frac{d}{dt} \int f(\underbrace{tg(x_0) + (1-t)g(x_1)}_{\text{Manifold Mixup}})dP_0(x_0)dP_1(x_1)dt \right|,$$

which results in the following smoothness regularizer:

$$R_{\text{m-mixup}}^{\text{DP}}(f \circ g) = \int_0^1 \left| \int \langle \nabla_z f(tg(x_0) + (1-t)g(x_1)), g(x_0) - g(x_1) \rangle dP_0(x_0)dP_1(x_1) \right| dt.$$

Previous works (Bojanowski et al., 2018; Berthelot et al., 2018) have showed that interpolations between a pair of latent features correspond to semantically meaningful, smooth interpolations in input space. By constructing a curve in the latent space, we can better capture the semantic changes while traveling from $P_0$ to $P_1$.

**Extensions and Implementation** The derivations presented so far, can be easily extended to Equalized Odds (EO). In particular, Lemma 1 can be extended to $\Delta$EO by interpolating $P_0^y$ and $P_1^y$ for $y \in \{0, 1\}$:

$$\Delta\text{EO}(f) = \sum_{y \in \{0,1\}} \left| \int_0^1 \frac{d}{dt} \int f(T(x_0, x_1, t))dP_0^y(x_0)dP_1^y(x_1)dt \right|.$$

The corresponding mixup regularizers can be obtained similarly by substituting $P_0$ and $P_1$ in $R_{\text{mixup}}$ and $R_{\text{m-mixup}}$ with $P_0^y$ and $P_1^y$:

$$R_{\text{mixup}}^{\text{EO}}(f) = \sum_{y \in \{0,1\}} \int_0^1 \left| \int \langle \nabla_x f(tx_0 + (1-t)x_1), x_0 - x_1 \rangle dP_0^y(x_0)dP_1^y(x_1) \right| dt.$$

Our formulation also encompasses other fairness metrics that quantify the *expected difference* between groups. This includes group fairness metrics such as accuracy equality which compares the mistreatment rate between groups (Berk et al., 2018). Similar to equation (1), we formulate a penalized optimization problem to enforce fairness via *fair mixup*:

$$\text{(Fair Mixup):} \quad \min_f \mathbb{E}_{(x,y)\sim P}[\ell(f(x), y)] + \lambda R_{\text{mixup}}(f). \tag{4}$$

Implementation-wise, we follow Zhang et al. (2018b) where only one $t$ is sampled per batch to perform mixup. This strategy works well in practice and reduce the computational requirements.

## 5.1 THEORETICAL ANALYSIS

To gain deeper insight, we analyze the optimal solution of fair mixup in a simple case. In particular, we consider the classification loss $\ell(f(x), y) = -yf(x)$ and the following hypothesis class:

$$\mathcal{H} = \{f | f(x) = \langle v, \Phi(x) \rangle, v \in \mathbb{R}^m, \Phi : \mathcal{X} \to \mathbb{R}^m\}.$$

Define $m_\pm = \mathbb{E}_{x \sim \mathbb{P}_\pm} \Phi(x)$, the label conditional mean embeddings, and $m_0 = \mathbb{E}_{x \sim \mathbb{P}_0} \Phi(x)$ and $m_1 = \mathbb{E}_{x \sim \mathbb{P}_1} \Phi(x)$, the group mean embeddings. We then define the expected difference $\delta_\pm = m_+ - m_-$ and $\delta_{0,1} = m_0 - m_1$. To derive an interpretable solution, we will consider the L2 variants of the penalized optimization problem. The following proposition gives the analytical solution when we regularize the model with $\Delta$DP.

**Proposition 2. (Gap Regularization)** *Consider the following minimization problem*

$$\min_{f \in \mathcal{H}} \mathbb{E}_{(x,y)\sim P}[\ell(f(x), y)] + \frac{\lambda_1}{2}\Delta\text{DP}(f)^2 + \frac{\lambda_2}{2}||f||_{\mathcal{H}}^2.$$

*For a fixed embedding $\Phi$, the optimal solution $f^*$ corresponds to $v^*$ given by the following closed form:*

$$v^* = \frac{1}{\lambda_2}\left(\delta_\pm - proj_{\delta_{0,1}}^{\frac{\lambda_2}{\lambda_1}}(\delta_\pm)\right),$$

*where $proj$ is the soft projection defined as $proj_u^\beta(x) = \frac{u \otimes u}{||u||^2 + \beta}x$.*

The solution $v^*$ can be interpreted as the projection of the label discriminating direction $\delta_\pm$ on the subspace that is orthogonal to the group discriminating direction $\delta_{0,1}$. By projecting to this orthogonal subspace, we can prevent the model from using group specific directions, that are unfair directions when performing prediction. Interestingly, the projection trick has been used in Zhang et al. (2018a), where they subtract the gradient of the model parameters in each update step with its projection on unfair directions. We then prove the optimal solution of fair mixup with the same setup as above. Similarly, we introduce an L2 variant of the fair mixup regularizer defined as follows:

$$R_{\text{mixup}}^{\text{DP-2}}(f) = \int_0^1 \left| \int \langle \nabla_x f(tx_0 + (1-t)x_1), x_0 - x_1 \rangle \, dP_0(x_0) dP_1(x_1) \right|^2 dt,$$

where we consider the squared absolute value of the derivative within the integral, in order to get a closed form solution.

**Proposition 3. (Fair Mixup)** *Consider the following minimization problem*

$$\min_{f \in \mathcal{H}} \mathbb{E}_{(x,y) \sim P}[\ell(f(x), y)] + \frac{\lambda_1}{2} R_{\text{mixup}}^{\text{DP-2}}(f) + \frac{\lambda_2}{2} \|f\|_{\mathcal{H}}^2.$$

*Let $m_t = \mathbb{E}_{x_0 \sim P_0, x_1 \sim P_1}[\Phi(tx_0 + (1-t)x_1)]$ be the $t$ dependent mean embedding, and $\dot{m}_t$ its derivative with respect to $t$. Let $D$ be a positive-semi definite matrix defined as follows: $D = \int_0^1 \dot{m}_t \otimes \dot{m}_t dt$. Given an embedding $\Phi$, the optimal solution $v^*$ has the following form:*

$$v^* = (\lambda_1 D + \lambda_2 I_m)^{-1} \delta_\pm.$$

Hence the optimal fair mixup classifier can be finally written as :

$$f(x) = \left\langle \delta_\pm, (\lambda_1 D + \lambda_2 I_m)^{-1} \Phi(x) \right\rangle,$$

which means that fair mixup changes the geometry of the decision boundary via a new dot product in the feature space that ensures group fairness, instead of simply projecting on the subspace orthogonal to a single direction as in gap regularization. This dot product leads to a Mahalanobis distance in the feature space that is defined via the covariance of time derivatives of mean embeddings of intermediate densities between the groups. To understand this better, given two points $x_0$ in group 0 and $x_1$ in group 1, by the mean value theorem, there exists $x_c$ such that:

$$f(x_0) = f(x_1) + \langle \nabla f(x_c), x_0 - x_1 \rangle = f(x_1) + \left\langle \delta_\pm, (\lambda_1 D + \lambda_2 I)^{-1} J\Phi(x_c)(x_0 - x_1) \right\rangle$$

Note that $D$ provides the correct average conditioning for $J\Phi(x_c)(x_0 - x_1)$, this can be seen from the expression of $\dot{m}_t$ (D is a covariance of $J\Phi(x_c)(x_0 - x_1)$). This conditioned Jacobian ensures that the function does not fluctuate a lot between the groups, which matches our motivation.

## 6 EXPERIMENTS

We now examine *fair mixup* with binary classification tasks on tabular benchmarks (Adult), visual recognition (CelebA), and language dataset (Toxicity). For evaluation, we show the trade-offs between average precision (AP) and fairness metrics ($\Delta$DP/$\Delta$EO) by varying the hyper-parameter $\lambda$ in the objective. We evaluate both AP and fairness metrics on a testing set to assess the generalizability of learned models. For a fair comparison, we will compare *fair mixup* with baselines that optimize the fairness constraint at training time. In particular, we compare our method with (a) empirical risk minimization (ERM) that trains the model without regularization, (b) gap regularization, which directly regularizes the model as given in Equation (1), and (c) adversarial debiasing (Zhang et al., 2018a) introduced in section 2. Details about the baselines and experimental setups for each dataset can be found in appendix.

### 6.1 ADULT

UCI Adult dataset (Dua & Graff, 2017) contains information about over 40,000 individuals from the 1994 US Census. The task is to predict whether the income of a person is greater than $50k given attributes about the person. We consider gender as the sensitive attribute to measure the fairness of the algorithms. The models are two-layer ReLU networks with hidden size 200. We only evaluate input mixup for Adult dataset as the network is not deep enough to produce meaningful

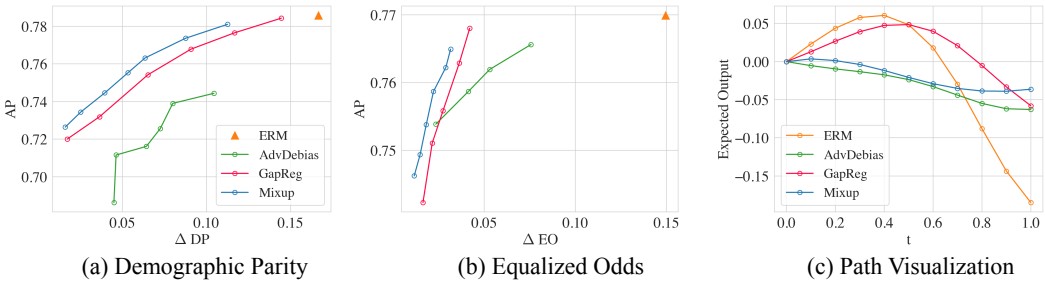

Figure 3: **Adult Dataset.** (a,b) The tradeoff between AP and $\Delta$DP/$\Delta$EO. (c) Visualization of the mixup path for models that regularize $\Delta$DP with different algorithms. We plot the calibrated curve $\mu'_f(t) := \mu_f(t) - \mu_f(0)$ for a better visualization. In this case, $\mu'_f(0) = 0$ and $|\mu'_f(1)| = \Delta$DP for all the calibrated curves $\mu'_f$. Therefore, we can compare the $\Delta$DP of each method with the absolute value of the last points ($t = 1$). The flatness of the path is highly correlated with the $\Delta$DP.

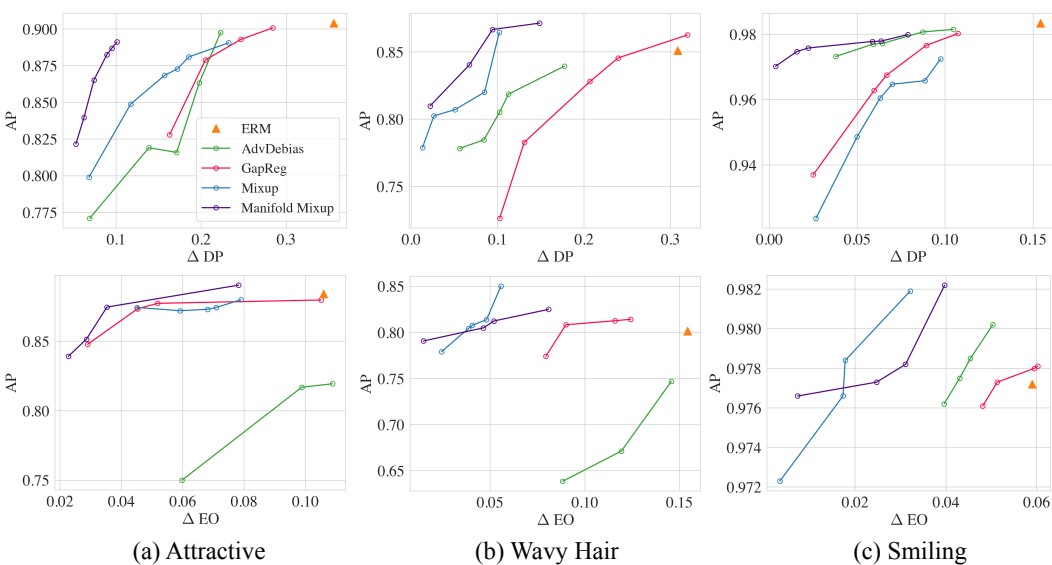

Figure 4: **CelebA Dataset.** The tradeoff between AP and $\Delta$DP/$\Delta$EO are shown in the first/second row for each task. Manifold mixup consistently outperforms the baseline across tasks.

latent representations. We retrain each model 10 times and report the mean accuracy and fairness measurement. In each trial, the dataset is randomly randomly split into a training, validation, and testing set with partition 60%, 20%, and 20%, respectively. The models are then selected via the performance on the validation set.

Figures 3 (a) shows the tradeoff between AP and $\Delta$DP. We can see that fair mixup consistently achieves a better tradeoff compared to the baselines. We then show the tradeoff between AP and $\Delta$EO in figure 3 (b). For this metric, fair mixup performs slightly better than directly regularizing the EO gap. Interestingly, fair mixup even achieves a better AP compared to ERM, indicating that mixup regularization not only improves the generalization of fairness constraints but also overall accuracy. To understand the effect of fair mixup, we visualize the expected output $\mu_f$ along the path for each method (i.e $\mu_f$ as function of $t$). For a fair comparison, we select the models that have similar AP for the visualization. As we can see in figure 3 (c), the flatness of the path is highly correlated to $\Delta$DP. Traininig without any regularization leads to the largest derivative along the path, which eventually leads to large $\Delta$DP. All the fairness-aware algorithms regularize the slope to some extent, nevertheless, fair mixup achieves the shortest arc length and hence leads to the smallest $\Delta$DP.

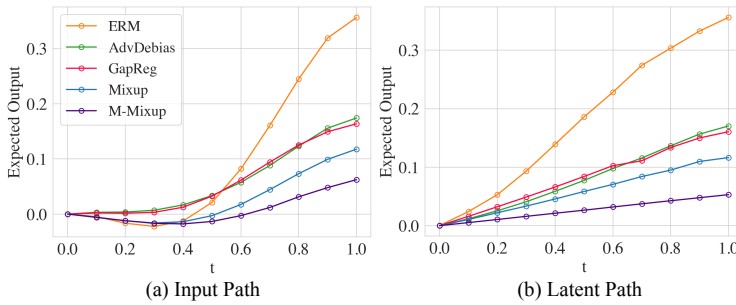

(a) Input Path  (b) Latent Path

Figure 5: **Visualization of calibrated paths** on attractive classification task for $\Delta$DP regularized models. The flatness of both input and latent path plays an important role in regularizing $\Delta$DP.

## 6.2 CELEBA

Next, we show that fair mixup generalizes well to high-dimensional tasks with the CelebA face attributes dataset (Liu et al.). CelebA contains over 200,000 images of celebrity faces, where each image is associated with 40 human-labeled binary attributes including gender. Among the attributes, we select attractive, smile, and wavy hair and use them to form three binary classification tasks while treating gender as the sensitive attribute[2]. The reason we choose these three attributes is that there exists in all these tasks, a sensitive group that has more positive samples than the other one. For each task, we train a ResNet-18 (He et al., 2016) along with two hidden layers for final prediction. To implement manifold fair mixup, we interpolate the representations before the average pooling layer.

The first row in figure 4 shows the tradeoff between AP and $\Delta$DP for each task. Again, fair mixup consistently outperforms the baselines by a large margin. We also observe that manifold mixup further boosts the performance for all the tasks. The tradeoffs between AP and $\Delta$EO are shown in the second row of figure 4. Again, both input mixup and manifold mixup yields well generalizing classifiers. To gain further insights, we plot the path in both input space and latent space in figure 5 (a) and (b) for the "attractive" attribute classification task. Fair mixup leads to a smoother path in both cases. Without mixup augmentation, gap regularization and adversarial debiasing present similar paths and both have larger $\Delta$DP. We also observe that the expected output $\mu_f$ in the latent path is almost linear with respect to the continuous sensitive attribute $t$, manifold mixup being the curve with the smallest slope and hence smallest $\Delta$DP.

## 6.3 TOXICITY CLASSIFICATION

Lastly, we consider comment toxicity classification with Jigsaw toxic comment dataset (Jigsaw, 2018). The data was initially released by Civil Comments platform, which was then extended to a public Kaggle challenge. The task is to predict whether a comment is toxic or not while being fair across groups. A subset of comments have been labeled with identity attributes, including gender and race. It has been shown that some of the identities (e.g., black) are correlated with the toxicity label. In this work, we consider race as the sensitive attribute and select the subset of comments that contain identities black or asian, as these two groups have the largest gap in terms of probability of being associated with a toxic comment. We use pretrained BERT embeddings (Devlin et al., 2019) to encode each comment into a vector of size 768. A three layer ReLU network is then trained to perform the prediction with the encoded feature. We directly adopt manifold mixup since input mixup is equivalent to manifold mixup by simply setting the encoder $g$ to BERT. Similarly, we retrain each model 10 times using randomly split training, validation, and testing sets, and report mean accuracy and fairness measurement.

Figures 6 (a) and (b) show the tradeoff between AP and $\Delta$DP/$\Delta$EO, respectively. Again, fair mixup consistently achieves a better tradeoff for both $\Delta$DP and $\Delta$EO. We then show the visualization of calibrated paths for $\Delta$DP-regularized models in Figure 6 (c). We can see that even with the powerful BERT embedding, all the baselines present fluctuated paths with similar patterns. In contrast, fair mixup introduces a nearly linear curve with a small slope, which eventually leads to the smallest $\Delta$DP.

---

[2]Disclaimer: the attractive experiment is an illustrative example and such classifiers of subjective attributes are not ethical.

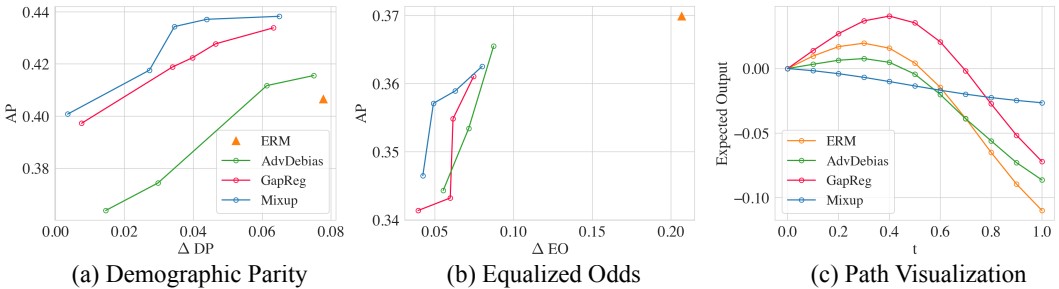

Figure 6: **Toxic Classification** (a,b) The tradeoff between AP and $\Delta$DP/$\Delta$EO. (c) Visualization of the calibrated paths for models that regularize $\Delta$DP with different algorithms. Interestingly, fair mixup presents a nearly linear curve with small slope, while the baselines introduce "inverted-U" shaped curves.

## 7 CONCLUSION

In this work, we propose *fair mixup*, a data augmentation strategy to optimize fairness constraints. By bridging sensitive groups with interpolated samples, fair mixup consistently improves the generalizability of fairness constraints across benchmarks with different modalities. Interesting future directions include (1) generating interpolated samples that lie on the natural data manifold with generative models or via dynamic optimal transport paths between the groups (Benamou & Brenier, 2000), (2) extending fair mixup to other group fairness metrics such as accuracy equality, and (3) estimating the generalization of fairness constraints (Chuang et al., 2020).

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

## A PROOFS

### A.1 PROOF OF LEMMA 1

**Lemma 1.** *Let $T : \mathcal{X}^2 \times [0,1] \rightarrow \mathcal{X}$ be a function continuously differentiable w.r.t. $t$ such that $T(x_0, x_1, 0) = x_0$ and $T(x_0, x_1, 1) = x_1$. For any differentiable function $f$, we have*

$$\Delta \mathrm{DP}(f) = \left| \int_0^1 \frac{d}{dt} \int f(\underbrace{T(x_0, x_1, t)}_{\text{interpolation}}) dP_0(x_0) dP_1(x_1) dt \right|.$$

*Proof.* The result follows from the fundamental theorem of calculus. In particular, given an interpolator $T$, we first rewrite the $\Delta$DP with the $T$:

$$\Delta \mathrm{DP}(f) = |\mathbb{E}_{x \sim P_0} f(x) - \mathbb{E}_{x \sim P_1} f(x)|$$
$$= |\mathbb{E}_{x_0 \sim P_0, x_1 \sim P_1} f(x_0) - f(x_1)|$$
$$= |\mathbb{E}_{x_0 \sim P_0, x_1 \sim P_1} f(T(x_0, x_1, 0)) - \mathbb{E}_{x_0 \sim P_0, x_1 \sim P_1} f(T(x_0, x_1, 1))|.$$

Not that $\mathbb{E}_{x_0 \sim P_0, x_1 \sim P_1} f(T(x_0, x_1, t))$ is a real-valued continuous function on $t \in [0, 1]$. Therefore, we have the following equivalence via the fundamental theorem of calculus:

$$\Delta \mathrm{DP}(f) = |\mathbb{E}_{x_0 \sim P_0, x_1 \sim P_1} f(T(x_0, x_1, 0)) - \mathbb{E}_{x_0 \sim P_0, x_1 \sim P_1} f(T(x_0, x_1, 1))|.$$
$$= \left| \int_0^1 \frac{d}{dt} \mathbb{E}_{x_0 \sim P_0, x_1 \sim P_1} f(T(x_0, x_1, t)) \right|.$$

$\square$

### A.2 PROOF OF PROPOSITION 2

**Proposition 2. (Gap Regularization)** *Consider the following minimization problem*

$$\min_{f \in \mathcal{H}} \mathbb{E}_{(x,y) \sim P}[\ell(f(x), y)] + \frac{\lambda_1}{2} \Delta \mathrm{DP}(f)^2 + \frac{\lambda_2}{2} \|f\|_{\mathcal{H}}^2.$$

*For a fixed embedding $\Phi$, the optimal solution $f^*$ corresponds to $v^*$ given by following closed form:*

$$v^* = \frac{1}{\lambda_2} \left( \delta_{\pm} - proj_{\delta_{0,1}}^{\frac{\lambda_2}{\lambda_1}}(\delta_{\pm}) \right),$$

*where $proj$ is the soft projection defined as $proj_u^\beta(x) = \frac{u \otimes u}{\|u\|^2 + \beta} x$.*

*Proof.* The problem above can be written as follows:

$$\min_{v \in \mathbb{R}^m} \mathcal{L}(v) := -(\langle v, \delta_{\pm} \rangle) + \frac{\lambda_1}{2} |\langle v, \delta_{0,1} \rangle|^2 + \frac{\lambda_2}{2} \|v\|_2^2$$

Setting first order condition to zero

$$\nabla_v \mathcal{L}(v) = -\delta_{\pm} + \lambda_1 \delta_{0,1} \otimes \delta_{0,1} v + \lambda_2 v = 0,$$

we obtain

$$(\lambda_1 \delta_{0,1} \otimes \delta_{0,1} + \lambda_2 I_m) v^* = \delta_{\pm}.$$

By inverting and applying the Sherman-Morrison Lemma, we have

$$\begin{aligned}
v^* &= (\lambda_1 \delta_{0,1} \otimes \delta_{0,1} + \lambda_2 I_m)^{-1} \delta_{\pm} \\
&= \lambda_1^{-1} \left( \frac{\lambda_2}{\lambda_1} I_m + \delta_{0,1} \otimes \delta_{0,1} \right)^{-1} \delta_{\pm} \\
&= \lambda_1^{-1} \left( I_m - \frac{(\frac{\lambda_1}{\lambda_2})^2 \delta_{0,1} \otimes \delta_{0,1}}{1 + \|\delta_{0,1}\|^2 \frac{\lambda_1}{\lambda_2}} \right) \delta_{\pm} \\
&= \frac{1}{\lambda_2} \left( I_m - \frac{\delta_{0,1} \otimes \delta_{0,1}}{\frac{\lambda_2}{\lambda_1} + \|\delta_{0,1}\|^2} \right) \delta_{\pm}.
\end{aligned}$$

Note the soft projection on a vector $u$ is defined as follows:

$$proj_u^\beta(x) = \frac{u \otimes u}{||u||^2 + \beta}x.$$

It follows that

$$v^* = \frac{1}{\lambda_2}\left(\delta_\pm - proj_{\delta_{0,1}}^{\frac{\lambda_2}{\lambda_1}}(\delta_\pm)\right),$$

which can be interpreted as the projection of the label discriminating direction $\delta_\pm$ on the subspace that is orthogonal to the group discriminating direction $\delta_{0,1}$. $\square$

### A.3 Proof of Proposition 3

**Proposition 3. (Fair Mixup)** *Consider the following minimization problem*

$$\min_{f \in \mathcal{H}} \mathbb{E}_{(x,y)\sim P}[\ell(f(x), y)] + \frac{\lambda_1}{2}R_{\text{mixup}}^{\text{DP-2}}(f) + \frac{\lambda_2}{2}||f||_{\mathcal{H}}^2.$$

*Define $m_t = \mathbb{E}_{x_0 \sim P_0, x_1 \sim P_1}[\Phi(tx_0 + (1-t)x_1)]$ be the t dependent mean embedding, and $\dot{m}_t$ its derivative with respect to t. Let $D$ be a positive-semi definite matrix defined as follows: $D = \int_0^1 \dot{m}_t \otimes \dot{m}_t dt$. Given an embedding $\Phi$, the optimal solution $v^*$ has the following form:*

$$v^* = (\lambda_1 D + \lambda_2 I_m)^{-1}\delta_\pm.$$

*Proof.* For $t \in [0, 1]$, we note by $\rho_t$ the distribution of $T(x_0, x_1, t)$, for $x_0 \sim P_0, x_1 \sim P_1$, and note $m_t = \mathbb{E}_{x \sim \rho_t}\Phi(x)$, and $\dot{m}_t$ its time derivative. We consider the $\ell_2$ variant of $R_T(f)$ in the analysis:

$$
\begin{aligned}
R_T^2(f) &= \int_0^1 \left|\frac{d}{dt}\mu_f(t)\right|^2 dt = \int_0^1 \left|\frac{d}{dt}\langle v, m_t\rangle\right|^2 dt \\
&= \int_0^1 |\langle v, \dot{m}_t\rangle|^2 dt = \left\langle v, \left(\int_0^1 \dot{m}_t \otimes \dot{m}_t dt\right)v\right\rangle,
\end{aligned}
$$

We then expand $\dot{m}_t$ when $T$ is mixup:

$$\dot{m}_t = \frac{d}{dt}\mathbb{E}_{x_0 \sim \mathbb{P}_0, x_1 \sim \mathbb{P}_1}\Phi(tx_0 + (1-t)x_1) = \mathbb{E}_{x_0 \sim \mathbb{P}_0, x_1 \sim \mathbb{P}_1}J\Phi(tx_0 + (1-t)x_1)(x_0 - x_1)$$

where $J$ denotes the Jacobian. Note that $D = \int_0^1 \dot{m}_t \otimes \dot{m}_t dt$, hence for the classification with fair mixup regularizer, the problem is equivalent to:

$$\min_{v \in \mathbb{R}^m} \mathcal{L}(v) := -(\langle v, \delta_\pm\rangle) + \frac{\lambda_1}{2}\langle v, Dv\rangle + \frac{\lambda_2}{2}||v||_2^2$$

Setting first order condition we obtain $(\lambda_1 D + \lambda_2 I_m)v^* = \delta_\pm$, which gives the optimal solution

$$v^* = (\lambda_1 D + \lambda_2 I_m)^{-1}\delta_\pm.$$

The corresponding optimal fair mixup classifier can be finally written as

$$f(x) = \langle\delta_\pm, (\lambda_1 D + \lambda_2 I_m)^{-1}\Phi(x)\rangle.$$

$\square$

## B Experiment Details

**Adult** We follow the preprocessing procedure of Yurochkin et al. (2019) by removing some features in the dataset[3]. We then encode the discrete and quantized continuous attributes with one-hot encoding. We retrain each model 10 times with batch size 1000 and report the mean accuracy and fairness measurement. The models are selected via the performance on validation set. In each trial, the dataset is randomly split into training and testing set with partition 80% and 20%, respectively. The models are optimized with Adam optimizer (Kingma & Ba, 2014) with learning rate $1 \times e^{-3}$. For DP, we sample 500 datapoints for each $A \in \{0, 1\}$ to form a batch. Similarly, for EO, we sample 250 datapoints for each $(A, Y)$ pair where $A, Y \in \{0, 1\}$.

---

[3]https://github.com/IBM/sensitive-subspace-robustness

**CelebA** Model-wise, we extract the feature of size $512$ after the average pooling layer of ResNet-18. A two-layer ReLU network with hidden size $512$ is then trained to perform prediction. Percentage of positive-labeled datapoints for attractive, wavy hair, and smiling that is male are $22.7\%$, $18.36\%$, and $34.6\%$, respectively. We use the original validation set of CelebA to perform model selection and report the accuracy and fairness metrics on the testing set. The visualization paths are also plotted with respect to the testing data. To implement manifold mixup, we interpolate the spatial features before the average pooling layer. Similarly, all the models are optimized with Adam optimizer with learning rate $1 \times e^{-3}$.

**Toxicity Classification** We download the Jigsaw toxic comment dataset from Kaggle website[4]. Percentage of positive-labeled datapoints for black and asian are $18.8\%$ and $6.4\%$, respectively, which together results in a dataset of size $22835$. We retrain each model $10$ times with batch size $200$ and report the mean accuracy and fairness measurement. The models are selected via the performance on validation set. The batch-sampling and data splitting procedure is the same as the one for Adult dataset. The models are again optimized with Adam optimizer with learning rate $1 \times e^{-3}$.

## C    ADDITIONAL EXPERIMENTS

### C.1    TRAINING PERFORMANCE

In Figure 7, we show the training performance for Adult dataset. As expected, GapReg outperforms Fair mixup on the training set since it directly optimizes the fairness metric. The results also support our motivation: the constraints that are satisfied during training might not generalize at evaluation time.

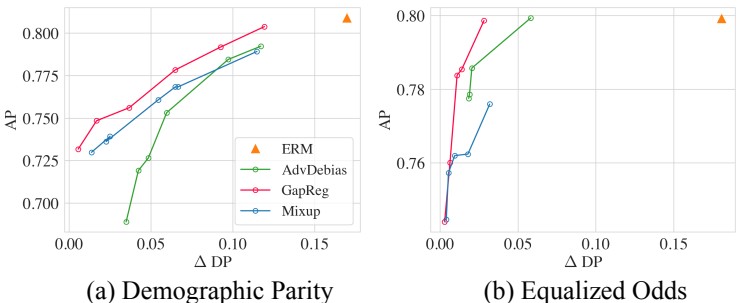

(a) Demographic Parity                    (b) Equalized Odds

Figure 7: **Training Performance on Adult Dataset.** The tradeoff between AP and $\Delta$DP/$\Delta$EO on training set.

### C.2    EVALUATION METRIC

The relaxed evaluation metric could overestimate the performance when the predicted confidence is significantly different between groups. For instance, a classifier $f$ can be completely unfair while satisfying this condition: $f(x) = 1$ w.p. $60\%$, $0$ w.p. $40\%$ on $P_0$, and $f(x) = 0.6$ w.p. $100\%$ on $P_1$. This satisfies this expectation-based condition. However, it is highly unfair if we binarize the prediction by setting the threshold $= 0.5$.

To overcome this issue, let $f_t$ be the binarized predictor $f_t(x) = \mathbb{1}(f(x) \geq t)$, we evaluate the model with average $\Delta$DP ($\overline{\Delta\text{DP}}$) defined as follows:

$$\overline{\Delta\text{DP}}(f) = \frac{1}{|\mathcal{T}|} \sum_{t \in \mathcal{T}} |\mathbb{E}_{x \sim P_0} f_t(x) - \mathbb{E}_{x \sim P_1} f_t(x)| \, ;$$

$$\overline{\Delta\text{EO}}(f) = \frac{1}{|\mathcal{T}|} \sum_{t \in \mathcal{T}} \sum_{y \in \{0,1\}} \left| \mathbb{E}_{x \sim P_0^y} f_t(x) - \mathbb{E}_{x \sim P_1^y} f_t(x) \right| ,$$

---

[4]https://www.kaggle.com/c/jigsaw-unintended-bias-in-toxicity-classification/data

where $\mathcal{T}$ is a set of threshold values. $\overline{\Delta\text{DP}}$ averages the $\Delta\text{DP}$ with binarized predictions derived via different thresholds. For instance, by averaging the $\Delta\text{DP}$ with thersholds $\mathcal{T} = [0.1, 0.2, \cdots., 0.9]$, $\overline{\Delta\text{DP}} = 0.5$ instead of $0$ for the example above, which captures the unfairness between groups. We report $\overline{\Delta\text{DP}}$ for each methods with thersholds $\mathcal{T} = [0.1, 0.2, \cdots., 0.9]$ in Figure 8. Similarly, fair mixup exhibits the best tradeoff comparing to the baselines for demographic parity. For equalized odds, the performances of fair mixup and GapReg are similar, where fair mixup achieves a better tradeoff when $\overline{\Delta\text{EO}}$ is small.

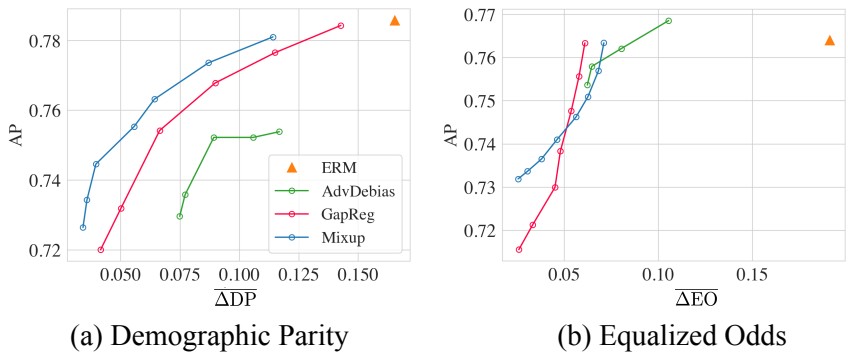

(a) Demographic Parity          (b) Equalized Odds

Figure 8: **Average $\Delta$DP and $\Delta$EO on Adult Dataset.**

## C.3 SMALLER MODEL SIZE

To examine the effect of model size, we reduce the hidden size from $200$ to $50$ and show the result in Figure 9. Overall, the performance does not vary significantly after reducing the model size. We can again observe that fair mixup outperform the baselines for $\Delta$DP. Similar to the results in section C.2, the performances of fair mixup and GapReg are similar, where fair mixup achieves a better tradeoff when $\Delta$EO is small.

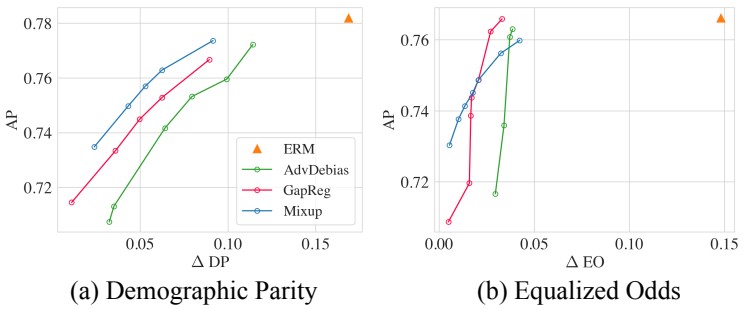

(a) Demographic Parity          (b) Equalized Odds

Figure 9: **Reducing the Model Size on Adult Dataset.**

