# OpenReview forum: "Fair Mixup: Fairness via Interpolation"
_ICLR.cc/2021/Conference — ICLR 2021 Poster_

### Official Review · AnonReviewer2 · 2020-10-23

**Rating:** 7
**Confidence:** 4

**Review:**

Brief Summary:

This paper introduces a method to improve fairness generalization.  The authors utilize a modified version of mixup where they regularize in favor of invariance of classifier prediction for interpolations between sensitive groups.  The idea is that we expect changes to sensitive attributes not to affect classification decisions.  They introduce a number of regularization terms for both demographic parity and equal opportunity.  Because the mixup interpolations are initially taken as linear interpolations between data --- and thus may not be naturally occurring --- they also introduce a version where samples are taken from a latent space.  They perform assessments on a number of datasets and modalities including Adult, CelebA, and  Jigsaw toxic comments.  They find their methods improve fairness generalization compared to a few baselines, including directly regularizing the model for fairness.

Questions + Comments:

Section 1:
- Figure 1 is a nice example. It could be good if Expected Output was written out a bit more explicitly or make it so that $\Delta DP$ is clearly the difference between the two points.  When I glanced at this graph at first, I thought $\Delta DP$ was being plotted --- consider that readers haven't reached the methods section at this point.  I was somewhat confused about what was being shown at first.

Section 4:
- When introducing lemma 1, I'd recommend stating explicitly that $T(x_0, x_1, t)$ corresponds to a function that outputs an interpolated sample between $x_0$ and $x_1$ at step $t$. This is clarified immediately below the lemma, but currently, the reader is left guessing when they first read the lemma as to what $T(x_0, x_1, t)$.  I found it slightly confusing while reading and ask the authors to consider stating it upfront.
- Why is  $P_0 (x_0)$ what we're integrating over? What is this notation meant to say? I notice that this is equivalent to the expected value of drawing $x_0 \sim P_0$, so is this meant to indicate the likelihood of each $x_0$ as well?

Section 5:
- When describing manifold mixup, there's an assumption that interpolations in the latent space $\mathcal{Z}$ will map nicely onto the set of legitimate data instances. If the encoder-decoder pair is an autoencoder trained without any specific regularization, it could be the case that many instances for linear interpolations could be off manifold.  I don't think this assumption is poor by any means, but I think describing the desired properties of the latent space could be a nice addition.

Section 6:

- Subsection 6.1: The two layer ReLU model with hidden size of $200$ is very large for the adult data set.  Is it necessary to use such a large model for this method and train for $1000$ iterations, which also feels like a long time for such a small data set? Its surprising that mixup improves overall performance on the data so noticeably, and I'm wondering if the ERM model has overfit the data at this point --- though such training regimes may be needed for fair mixup given that only one $t$ is sampled iteration. Could the authors try something like (1) decreasing the training time or size of the model or (2) just add another baseline such as a random forest or logistic regression to this graph? I think this would help us understand the effects of fair mixup a bit better.

- Subsections 6.2: For selecting the models, did you always choose the model with best validation accuracy for both ERM and fairmixup? Further how long did you train?

- Subsection 6.3: Was model selection done in this experiment?

Overall Experimental questions:

- I'm curious as to the training requirements imposed by the use of this method. The authors compare ERM models trained at as many iterations as the fairmixup models.  However, given that only one $t$ is sampled per batch, it might take quite a few iterations to get a good fairmixup model.  Is this the correct intuition? Will the ERM models produce optimal solutions long before fairmixup? Further, if we increase the number of $t$'s, will the fairmixup models converge quicker at the cost of computation time? I'd appreciate if the authors add a bit of discussion related to this.  If the question is unfounded, I'd appreciate clarification.

For the experimental selection, some model selection for the adult data sets and toxicity classification would add the experiments and help disentangle the effects of overfitting (unless I'm missing something) -- particularly in the case of the adult data set because right now, I'm inclined to think the model is overfitting.

Overall, I'm fairly convinced this method is working well and appreciate the breadth of data modalities and experiments the authors performed performed.

Overall thoughts:

I think this is a nice paper with a useful contribution to fairness generalization.  If it is really the case that this method is flexible enough to improve both fairness and classification generalization, this method could be quite useful in general. I have a couple clarification questions from the body of the paper, and I'd appreciate answers from the authors.  Further, I'd also appreciate it if the authors provided clarification to the experimental decisions I asked about above.

One last point for the authors to consider is that for CelebA, the authors build classifiers to preform attractiveness classification. Though they do so from the perspective of discouraging classifiers to exhibit biases, it still could be worthwhile to include a brief discussion in the appendix on the ethical considerations of such models.

---

> ### Author Response · Authors · 2020-11-16
> **Thank you for your review**
>
> Thank you for your helpful suggestions. We would like to address your questions as follows:
>
> Section 1:
>
> We agree that this figure is a bit confusing at first. We’ve replaced the expected output with expected parity (since the normalized expected output is equivalent to E1[f] - E0[f]), where we hope this will provide a better comparison between the unregularized and regularized approaches.
>
> Section 4:
>
> We’ve clarified the interpolator T before Lemma 1. Yes, integrating over P0 and P1 means that we are drawing samples from x0 \~ P0 and x1 \~ P1 to calculate the expected value, where we abuse the notation by using P0 and P1 as probability density functions when writing the integral.
>
> Section 5:
>
> We will clarify the assumption on the latent space. Since we are performing discriminative learning, it is hard to provide guarantee on the property of the latent space. It would be an interesting future direction to combine our approach with generative models such as VAE.
>
> Section 6.1:
>
> 1. To improve model selections, we rerun the adult experiments with a validation set splitted from the training set, which results in standard 60%/20%/20% train/val/test splits. While the overall performance is slightly improved with a validation set, fair mixup still consistently outperforms the baselines. We also observe that the optimal iteration for fair mixup and GapReg is around 400 iterations. This indicates that even if one t is sampled per iteration, fair mixup converges at the same speed as GapReg empirically. In contrast, AdvDebias takes roughly 1100 iterations to converge, where the instability of adversarial training might be the reason for the slower rate.
>
> 2. To examine the effect of model size, we reduce the hidden size from 200 to 50 and show the results in Appendix C.4. We can see that fair mixup still consistently outperforms the baselines when we train smaller models.
>
> Section 6.2:
>
> Yes, we train the model for 70 epochs in total and select the model with best validation accuracy or fairness metrics.
>
> Section 6.3:
> Similar to section 6.1, we retrain each model 10 times for fixed iterations (1000 iters) and report the averaged performance. We will add the experiment results with early stopping as we show for the adult dataset.
>
> Training Requirement and Convergence:
>
> After using validation set for model selection, we observe that ERM indeed produces optimal solutions before fair mixup and GapReg for Adult ddataset. Nevertheless, the performance of ERM is still not satisfied compared to the regularized approaches. The reason we use the same iterations is to follow the implementation of previous works., eg., Yurochkin et al [1]. After increasing the mixup size t per iteration, we observe very similar performance compared to our current implementation, in line with the observation from the original mixup paper [2].
>
> Celeb-A Classification:
> We agree that attractiveness is a sensitive attribute. The reason we select it is because we would like to demonstrate that the bias could exist when we encounter “subjective” attributes. For instance, we might have ground truth for wavy hair, but it is not ethical to label the attractiveness of a person. We add a disclaimer in the footnote that states that this example is an illustrative example and such classifiers of subjective attributes are not ethical.
>
> Thank you again for your suggestions.
>
> [1] Yurochkin et al., Training individually fair ML models with sensitive subspace robustness, https://github.com/IBM/sensitive-subspace-robustness, ICLR 2020.
> [2] Zhang et al., mixup: Beyond Empirical Risk Minimization, ICLR 2018.
>
> Thanks,
> Authors

---

> > ### Comment · AnonReviewer2 · 2020-11-16
> > **Response**
> >
> > Thanks for the updates. I looked through the new paper updates and appreciate the expansion on the adult data set experiments as well as some of the minor corrections / extensions.  The new adult data set results are useful extensions. After reading these and my co-reviewers comments, I still advocate for acceptance.

---

### Official Review · AnonReviewer3 · 2020-10-28
**Nice, clear paper, setup & experiments**

**Rating:** 7
**Confidence:** 4

**Review:**

Summary: This paper proposed a data augmentation method for training a classifier which is intended to have predictive parity between two identified groups. It is based on the “mixup” idea – samples from the two groups are interpolated between, and the smoothness of this path is encouraged. The authors recommend doing the interpolation in latent space. An optimal solution in a constrained setting is derived, and experiments show the empirical success of the model at this task on 3 datasets.

Recommendation: I recommend acceptance of this paper. The application of mixup to this task is sensible, the paper is clearly written, and the theoretical and empirical work both seem solid.

Strengths:
-	Using mixup for a task like this seems reasonable. There is a literature on data augmentation methods for this task
-	Explanation of the mixup method is clear and the theoretical work seems good
-	The empirical results are pretty strong, beating each method at each task. Showing the tradeoff is good, this is the right type of diagram to use here
-	I like that this method is non-adversarial – easier to train and more reliable imo

Weaknesses + Clarifications:
-	The question of the latent variable model seems relevant and interesting. It seems that the mixup method is only as good as the model, and also the trained model might add its own biases to the classification task. It would be nice to see some discussion of this in the paper
-	I am surprised that mixup improves precision on the adult task. It would be good to see some exploration of this
-	For experiments, are all runs shown? Or just the Pareto fronts.
-	A number of hyperparameters (e.g. regularization) are not given
-	For all the latent path figures (eg Fig 3) why is the y value at x= 0 always 0? Is it normalized to this? Be clear in your description (or maybe I missed it)
-	I would be interested in seeing some further analysis on this model, perhaps using the interpolations themselves

---

> ### Author Response · Authors · 2020-11-16
> **Thank you for your review**
>
> Thank you for your helpful suggestions. We would like to address your questions as follows:
>
> 1. Experiments
>
> 1.1 To explore the results on the Adult dataset, we rerunned the adult experiments with model selection via a validation set and show it in Appendix C.1. While the overall performance is slightly improved with early stopping, fair mixup still consistently outperforms the baselines for both DP and EO.
>
> 1.2 To remain simple, we only show the Pareto fronts.
>
> 1.3 Details of the hyperparameter are included in Appendix B for each dataset. We’ve added Table 1 to show the trade-off hyperparameters we use for each experiment.
>
> 1.4 More ablation studies of the model are added in Appendix C, which shall provide a more comprehensive picture of our approach.
>
> 2. Latent Variable Models
>
> 2.1 All the latent paths shown in figures are normalized by subtracting the mu(0) as we explained in the caption of Figure 3.
>
> 2.2 Thank you for your interest in our latent framework. Could you please clarify the first question with regards to latent variable models?
>
> Thanks,
> Authors

---

> > ### Comment · AnonReviewer3 · 2020-11-18
> > **Response**
> >
> > 1.1 - Thanks for running these. In my opinion this sort of selection with validation is a more principled way to do this experiment, and the figures in C.1 should maybe replace the ones in the main body
> >
> > 2.2 - Here is the question: different latent variable models may produce different paths between the groups. Might these types of differences occur in practice and would that impact the effectiveness of fair mixup?

---

> > > ### Author Response · Authors · 2020-11-19
> > > **Response**
> > >
> > > Thank you for your clarification. Yes, different latent paths could affect the tradeoff if the embedding g is not jointly learned. For instance, if we replace BERT embedding with any other sentence embedding, the accuracy of the predictor will take a hit if the embedding was not good, but fair mixup will maintain the fairness between groups. It is also an interesting future direction to combine our approach with other latent variable models such as VAE.
> > >
> > > Thanks,
> > > Authors

---

### Official Review · AnonReviewer4 · 2020-10-28
**An interesting topic but the theoretical results are not strong enough**

**Rating:** 6
**Confidence:** 4

**Review:**

This paper provided a fair mixup strategy to improve the generalizability of fair classifiers. The authors also provide theoretical understanding of the proposed methods. Extensive experimental results are presented to verify the effectiveness of proposed method. Mainly, I have two questions that need to be answered.

1) The authors claimed that "the proposed fair mixup can improve the generalization of group fairness metrics", and they "provide a theoretical analysis to deepen the understanding of the proposed method". However, it can not be seen from either proposition 1 or proposition 2 to show the improvement of generalization. Either proposition 1 or proposition 2 does not show the generalization bound. Could you please give some explanations?

2) In the theoretical analysis, the authors considered $f(x)$ as a linear function of $\Phi(x)$ where $\Phi$ is an embedding function. However, this is not true in many applications. Even for the experiments in this paper, this is not true. For example, two-layer ReLu networks (this is not a linear function) is used in section 6.1. If so, the theoretical results can not fully deepen the understanding of the proposed method, which makes the results less important. On the other hand, if general non-convex function $f(x)$ (for example, NN with ReLu activation function) is considered, then the optimal $v^*$ is very hard to obtain since this is a NP-hard for general non-convex minimization problem.

Finally, I am wondering if the proposed methods can be extended to multi-class classification since this paper is studying binary classification problem?

---

> ### Author Response · Authors · 2020-11-16
> **Thank you for your review**
>
> Thank you for your constructive comments. Below we would like to address your concerns.
>
> 1. Generalization Bound and Theoretical Analysis
>
> We did not claim the theory explains the generalization. Proposition 1 and 2 are proposed to show how fair mixup compares to GP minimization in terms of optimal function in a simple setting. The goal is to better understand the effect of the regularization on the optimal solutions. We agree that the theory does not hold for the general case and we did not claim it as our main contribution. It was developed to better understand the method in a simple case where a closed form solution exists. To make it clearer, we’ve clarified this at the beginning of section 5.1.
>
> 2. Extension to Multi-class
>
> Yes, we believed that it can be extended to multi-class framework by defining proper fairness metrics. For instance, we can define the DP and EO gap as the sum of the group difference over classes. Therefore, we can again regularize the smoothness of the path for each class. This can be done efficiently by sharing the computation of Jacobian when we use the standard multi-head classifiers.
>
> Thank you again for your suggestions.
>
> Thanks,
> Authors

---

### Official Review · AnonReviewer1 · 2020-10-29

**Rating:** 5
**Confidence:** 4

**Review:**

This paper proposes "fair mixup" for training fair classifiers.  Inspired by the mixup algorithm, which was presented to improve the generalization performance in Zhang et al., 2018b, fair mixup pick two samples from two different sensitive groups.  Instead of regularizing the gap (e.g., \delta DP), the authors regularized the derivative along the path between two samples.  This algorithm can be applied either to the input space or to the feature space.  The idea of using mixup for training fair classifiers is interesting, and the paper is well written and easy to read.  However, I have a few critical concerns.

Concerns:
- E_{x from P0} f(x) = E_{x from P1} f(x) does not imply fairness:  My first concern is the choice of this relaxed metric, which the authors borrowed from Madras et al. (2018).  Most surrogate conditions are sufficient conditions for the target fairness condition.  For instance, zero mutual information between the sensitive attribute and the classification output implies demographic parity.  The empirical covariance, used in Zafar et al., is also similar.  However, this one is neither necessary nor sufficient.  A classifier can be completely unfair while satisfying this condition: f(x) = 1 w.p. 60%, 0 w.p. 40% on P0, and f(x) = 0.6 w.p. 100% on P1.  This satisfies this expectation-based condition.  However, it is highly unfair against P0 as P(Y=1|P0) = 0.6 and P(Y=1|P1) = 1, assuming the threshold is 0.5.  I would make sense if this metric helps impose the actual fairness conditions we care about. Still, the authors only reported \Delta DP and \Delta EO, without reporting the actual DP/EO differences or multiplicative gaps.

- Does fairness condition not generalize?:  First of all, I haven't seen large gaps between the level of fairness measured in the train set and the test set.  I believe that it really depends on the choice of training algorithm, and most of the methods I have seen in the literature do not exhibit huge gaps.  Even when there is such a gap, one can choose the best model based on the validation performance (accuracy/fairness).  The authors may want to add more evidence on the lack of fairness generalization of existing algorithms to justify the considered problem.

- Other baseline algorithms with early stopping:  The authors seem not to use any validation or early-stoping to maximize the test performance.  This is commonly done in most work in the literature, so please clarify this or do so.  Most importantly, related to the above concern, GapReg's performance might be improved if used with proper validation and early stopping.  Also, please report the train performance of GapReg.  Its train performance should be better than Fair mixup in every aspect.  AdvDebias is not a good baseline algorithm as it's not optimized for a target task.  Please add proper baseline algorithms.

===

post-rebuttal:  The authors have addressed some of my concerns, but the experimental results are still missing several important baselines.   Raising my score from 4 to 5.

---

> ### Author Response · Authors · 2020-11-16
> **Thank you for your review**
>
> Thank you for your constructive comments. Below we would like to address your concerns.
>
> 1. Early Stopping and New Baseline
>
> As we state in appendix B, the validation set has already been used to perform model selection for Celeb-A. To improve model selections, we rerun the adult experiments with a validation set splitted from the training set, which results in standard 60%/20%/20% train/val/test splits. We implement LAFTR [1], which is the approach proposed to minimize DP and EO gaps. For fair comparison, we jointly optimize the discriminator when training the predictor and discard the reconstruction loss, while the original paper introduced a two stage training procedure. The results are shown in Appendix C.1. While the overall performance is slightly improved with early stopping, fair mixup still consistently outperforms the baselines for both DP and EO.
>
> 2. Training Performance and Generalization
>
> We show the training performance in Appendix C.2. As expected, GapReg outperforms Fair mixup on the training set since it directly optimizes the fairness metric. The results also support our motivation: the constraints that are satisfied during training might not generalize at evaluation time.
>
> There are previous works that discussed the generalization error of data-dependent constraints. For instance, Cotter et al [2] shows that the constraints *such as fairness* that are satisfied at training time might not behave the same at evaluation time for several datasets. Our experiments in 6.2, C.1, and C.2 also corroborate the existence of the generalization gap.
>
> 3. Fairness Metric
>
> We agree that the metric based on mean score could be problematic in some cases, nevertheless, mean score parity has been widely used in previous literatures [3][4][5]. To overcome the proposed issue, similar to average precision, we evaluate the model with average DP gap, which average the DP over binarized predictions derived via different thresholds. For instance, by averaging the DP with thresholds = [0.1, 0.2, …, 0.9], the average DP is 0.5 instead of 0 for the example above, which captures the unfairness between groups. For more details, please refer to the Appendix C.3. The experiment results show that fair mixup still outperforms the baselines in terms of the new metric (average DP) , indicating that it does not result in unfairly trivial solutions.
>
> Thank you again for your suggestions, we hope that our clarifications and new supportive experiments help the reviewer in reassessing the paper.
>
> [1] Madras et al., Learning Adversarially Fair and Transferable Representations, ICML 2018
> [2] Cotter et al., Training Well-Generalizing Classifiers for Fairness Metrics and Other Data-Dependent Constraints, ICML 2019
> [3] Costen et al., Fair transfer learning with missing protected attributes, AIES 2019
> [4] Wei et al., Optimized Score Transformation for Fair Classification, AISTATS 2020
> [5] Taskesen., A Distributionally Robust Approach to Fair Classification., 2020
>
> Thanks,
> Authors

---

> > ### Comment · AnonReviewer1 · 2020-11-16
> > **Re: performance on the Adult dataset**
> >
> > Thanks for your response.  Please find my follow-up questions below.
> >
> > ===
> >
> > 1. Looking at the new experimental results, I am still confused about the experimental results.
> > The numbers on the Adult dataset seem very different and much lower than what I have seen in the literature:
> >
> > See Fig. 1 of [https://dl.acm.org/doi/10.1145/3357384.3357974]
> > Note that their performance is also reported on the test set, as stated in Sec. 5.1.3.
> >
> > Also, the AdvDebias's performance on the Adult dataset was much better than what's reported in this submission.
> > See Table 3 and Table 4 in [Zhang et al.].
> >
> > Lastly, the performance of the most standard baseline algorithm is still not present: [https://arxiv.org/abs/1507.05259]
> > According to the first paper I mentioned above, this will also result in much higher performance than what's reported in this work.
> >
> > 2. Cotter et al. did report the generalization gap for a very specific method, which was proposed in their own previous work, but not for other algorithms.  Also, one quick question: what about the two-dataset approach proposed in Cotter et al.?  They seem to have important baselines to compare with.  Though the authors claimed that "game-theoretic approaches could be hard to scale for complex model classes", Cotter at al.'s paper also proposed a practical algorithm (Algorithm 4 in the paper), and showed that it performs well for their settings.

---

> > > ### Author Response · Authors · 2020-11-16
> > > **Re: performance on the Adult dataset**
> > >
> > > Thank you for your prompt response. We address your questions as follows:
> > >
> > > 1. The difference between the performance reported in our paper and the one reported in  papers the reviewer mentioned is due to different experimental setups considered, mainly: different (1) evaluation metrics and (2) models. We first note that these papers use metrics such as accuracy [1] or FPR/FNR [2]. In our work, we report the mean of average precision over 10 random splits. By setting the threshold to 0.5, all the approaches can achieve an accuracy greater than 0.8, in line with the results from previous works [1][3]. Secondly, for fair comparison, all of our experiments adopt the same multilayer neural network, while some of the previous works use different models such as linear SVM [3]. Different models or inductive bias will result in slightly different performances as we show in appendix C.4. While we do not directly compare to [3] due to model mismatch, we’ve included it in the related work section. To reproduce the experiment results for the Adult dataset, we also release our code in the supplementary material.
> > >
> > > 2. By comparing appendix C.1 and C.2, it is obvious that the generalization gap exists empirically. As it can be seen from appendix C.2 there is a clear overfitting on the training data of GP minimization. On the contrast fair mixup has lower performance on the training set (Appendix C.2), but it generalizes better on the test set (Appendix C.1).  Algorithm 4 in Cotter et al results in a stochastic classifier over T parameters where T is the number of game iterations. Therefore, it is not computationally feasible to extend it to larger datasets such as Celeb-A, where the final model would be an ensemble of T ResNets.
> > >
> > > Thank you again for your suggestions.
> > >
> > > [1] Iosifidis et al., AdaFair: Cumulative Fairness Adaptive Boosting
> > > [2] Zhang et al., Mitigating Unwanted Biases with Adversarial Learning
> > > [3] Zafar et al., Fairness Constraints: Mechanisms for Fair Classification
> > >
> > > Thanks,
> > > Authors

---

### Author Response · Authors · 2020-11-16
**Summary of changes in revision**

We thank all the reviewers for their helpful comments and feedback. We have updated our work based on the suggestions, which is summarized as follows:

1. Improve the clarification for Figure 1, Section 4, Section 5.1, and Section 6.2.

2. New empirical studies on early-stopping, training performance, fairness metrics, and model complexity for Adult dataset in Appendix C. These experiments will be extended for the other benchmarks.

All the modifications are marked with blue. We will address individual concerns and questions in the responses.

Thanks,
Authors

---

### Decision · Program_Chairs · 2021-01-07
**Final Decision**

**Decision:**

Accept (Poster)

**Comment:**

# Paper Summary

The goal of this paper is to improve generalization of fairness metrics by borrowing ideas from "mixup", which attempts to improve generalization in the non-fairness setting by introducing convex combinations of training examples as virtual examples.

They adapt this idea by interpolating between protected *groups*, and adding a regularizer that forces the classifier to vary smoothly along this interpolation path. To this end, they show that, for a particular interpolation function, the (empirical) disparity in the fairness metric is upper bounded by their proposed regularizer (which depends both on the fairness metric, and the interpolation function). They consider two fairness metrics (disparate impact and equalized odds) and two interpolation functions (convex combinations in the feature space, or in a latent space).

As Reviewer 4 points out, the above is not a complete explanation for why their regularizer works: they've only really shown that it upper bounds the empirical disparity in the fairness metric (and we could have regularized this empirical disparity directly, and indeed they do so, as a baseline, in their experiments). Presumably the intuition is that their regularizer is improving generalization by (implicitly) depending on virtual examples, but this isn't made explicit.

In a "theoretical analysis" section, they give closed form solutions using classification loss, along with L2 regularization and either (i) a regularizer penalizing the true disparity of impact or (ii) their proposed regularizer (which upper bounds the former). Both reviewer 4 and I seem to doubt if this adds much insight (the other reviewers didn't discuss this section).

They close with experiments on Adult, CelebA, and Jigsaw Toxicity, all of which show dramatic performance gains using their regularizer. However, they only compare to one external baseline (adversarial debiasing).

# Pros

1. Reviewers agreed that the paper was well-written
1. The derivation of their regularizer is somewhat complex, but is described step-by-step, and very clearly
1. Adapting mixup to the problem of improving fairness generalization seems natural and intuitive, but this intuition is maybe given short shrift in the later sections
1. Experiments show impressive results

# Cons

1. Reviewer 1 notes that having the expected value of the classification function be equal for both protected groups does not imply fairness, since the classification function would presumably be thresholded to make hard classification decisions
1. Reviewer 4 points out that they do not actually explain why their regularizer will improve generalization better than the "usual" disparity regularizer. Instead, they only show that it upper-bounds the empirical disparity in the fairness metric. Presumably, the intuition is that their mixup regularizer is doing something like adding "virtual samples"
1. I would like to see a more detailed explanation of how their regularizer is implemented, in the main text (they only say that it "can be easily optimized by computing the Jacobian of f on mixup samples")
1. Reviewers 1 and 2 would like more external baselines (there is only one at the moment, "adversarial robustness"), with reviewer 1 suggesting early stopping. The authors added a new early stopping experiment on CelebA to the appendix, but it would be nice to have this baseline included in all experiments in the main text

# Conclusion

Three of the four reviewers recommended acceptance, with the "reject" reviewer scoring it "5: weak reject". This reviewer had three main criticisms: (i) matching expected classification functions is not the same as matching classification *decisions*, (ii) fairness problems might not have a generalization problem to begin with, and (iii) the experiments don't include enough external baselines. I disagree with the second point, but agree with the other two. I think the third is the most critical, since the first could be solved in many cases by e.g. sampling instead of making hard deterministic decisions.

Overall, my opinion is that this is a borderline paper, but that it falls on the "accept" side of the boundary. The idea is intuitive, and exposition is clear, the derivation is quite interesting, and the experimental results are (aside from not having enough baselines) impressive.